

# Glacier changes and surges over Xinqingfeng and Malan Ice Caps in the inner Tibetan Plateau since 1970 derived from Remote Sensing Data

Zhen Zhang[1*], Shiyin Liu[2,3*], Zongli Jiang[4], Donghui Shangguan[3], Junfeng Wei[4], Wanqin Guo[3], Junli Xu[5], Yong Zhang[4] and Danni Huang[1]

[1]School of Geomatics, Anhui University of Science and Technology, Huainan, 232001, China

[2]Institute of International Rivers and Eco−Security, Yunnan University, Kunming, 650500, China

[3]State Key Laboratory of Cryospheric Science, Northwest Institute of Eco−Environment and Resources, Chinese Academy of Sciences, Lanzhou, 730000, China

[4]School of Resource Environment and Safety Engineering, Hunan University of Science and Technology, Xiangtan, 411201, China

[5]College of Urban and Environmental Sciences, Yancheng Teachers University, Yancheng, 224002, China

*Correspondence to*: Zhen Zhang (zhangzhen@aust.edu.cn) and Shiyin Liu (shiyin.liu@ynu.edu.cn)

**Abstract.** The inner Tibetan Plateau region is a glacierised area with heterogeneous variations. However, the detailed glacier area and mass changes in this region prior to the year 2000 are scarce, and there are limited processes available to understand this heterogeneity. In this paper, we present an integrated view of the glacier area and its mass changes for Mt. Xinqingfeng and Mt. Malan of the inner Tibetan Plateau as derived from topographic maps, Landsat, ASTER, SRTM DEM, and TerraSAR-X/TanDEM-X for the period of 1970-2012 and 1970-2018, respectively. Our results show that the glaciers experienced weak shrinkage in area by 0.09 ±0.03% from 1970 to 2018, but there was a median mass loss at a rate of 0.22 ± 0.17 m w.e. a$^{-1}$ and 0.29 ±0.17 m w.e. a$^{-1}$ during 1999-2012 in Mt. Xinqingfeng and Mt. Malan respectively. The glaciers of Mt. Malan have had a lower total mass loss of 0.19 ± 0.14 m w.e. a$^{-1}$ during 1970-1999. The mean velocity of the glaciers during 2013-2018 was 0.16 m d$^{-1}$, as demonstrated from the Global Land Ice Velocity Extraction from Landsat 8 (GoLIVE). The Monuomaha Glacier and Zu Glacier together with another 5 glaciers displayed the surging or advancing characteristics during the observation period. These glaciers showed a long active period of time and comparatively low velocities, which suggests that thermal controls are important for the surge initiation and recession. The ablation area or accumulation area exhibited small slopes with velocities that were too slow to remain in balance with the accumulation rate; thus, they required surging to transport mass from the reservoir area down the glacier tongue.

## 1 Introduction

Glaciers in the High Mountains Asia (HMA) are the headwater sources for many rivers and lakes. As a result, the HMA is often also known as the 'Asia Water Tower'(Immerzeel et al., 2010). In the recent warming climate, many sub-regions of the HMA, such as the Himalayas, Nyainqentanglha, Tien Shan, Bhutan, Nepal, and Spiti Lahaul, have experienced glacier mass



loss (Brun et al., 2017; Farinotti et al., 2015; Kääb et al., 2012; Kääb et al., 2015; Wu et al., 2018). However, glaciers in the Pamir, Karakoram and West Kunlun have on average been in near balance or have shown mass gain in recent years, although these results have been debated (Bao et al., 2015; Bolch et al., 2017; Gardelle et al., 2012b; Gardelle et al., 2013; Holzer et al., 2015; Kääb et al., 2015; Lin et al., 2017; Zhang et al., 2016). Glaciers in the Aru Co regions on the east of West

Kunlun, where two glaciers collapsed in 2016, also showed mass gains (+0.33 ± 0.61 m w.e. a$^{-1}$) in the early 21$^{st}$ century, which is in contrast to their mass loss (−0.15 ± 0.30 m w.e. a$^{-1}$) before 1999 (Zhang et al., 2018). A similar result was reported for the Kangzhag Ri at the centre of the inner Tibetan Plateau (TP) with a mass gain of +0.16 ± 0.02 m w.e. a$^{-1}$ after 1999 and a loss (−0.34 ± 0.01 m w.e. a$^{-1}$) before 1999 (Zhang and Liu, 2018). Glaciers in the Zangsar Kangri were reported as having a mass gain (+0.37 ± 0.25 m w.e. a$^{-1}$) during 2003-2009 (Neckel et al., 2014). These glaciers are mainly located in

the northwest and central parts of the Tibetan Plateau, and it is of great interest whether this same anomaly exists for glaciers that are further to the northeast of the plateau, for example, the Mt. Xinqingfeng and the Mt. Malan Ice Caps (collective known as XM). Although Zhou et al. (2019) reported Xinqingfeng and Malan mass budgets of −0.21 ± 0.10 m w.e. a$^{-1}$ and −0.22 ± 0.10 m w.e. a$^{-1}$, respectively, for the period of 2000-2016, the knowledge of the glacier mass change before 1999 remains unknown.

Geodetic mass estimates have revealed weak mass losses for the entire inner TP for the period following 2000. However, the inner TP region is an aggregation of glacier balance heterogeneous sub-regions (Brun et al., 2017; Neckel et al., 2014). Glaciers in the southern and southeastern parts of the TP have suffered from significant retreat, while glaciers in the western parts have remained relatively stable or have even advanced (Brun et al., 2017; Chen et al., 2017; Lin et al., 2017; Neckel et al., 2013; Xu et al., 2018; Zhang and Liu, 2018). The XM is located in the northeast of the inner TP, and detailed glacier

mass budget analyses for the XM glaciers, especially prior to the year 2000, are rare.

Glacier surging can cause hazards, such as related outburst floods and their associated impacts on the downstream areas (Kotlyakov et al., 2004; Motyka and Truffer, 2007). The HMA is one of the major 'superclusters' of glacier surge (Sevestre and Benn, 2015). However, most surged glacier clusters have been reported in Pamir, Karakoram and West Kunlun (Bhambri et al., 2017; Chudley and Willis, 2018; Copland et al., 2011; Kotlyakov et al., 2008; Shangguan et al., 2016). In

other regions, there have been sporadic reports of surging glaciers for the inner TP, such as, Aru Co, Ányêmaqên, Ulugh Muztagh, Namjagbarwa and Geladandong (Guo et al., 2013; Jiang et al., 2018; Xu et al., 2018; Zhang, 1983; Zhang et al., 2018). Although there is a high incidence and coverage of surge-type glaciers in the HMA, detailed analyses are rare and there could be more regions that contain surging glaciers that have not been unveiled. To our knowledge, there is a lack of reports regarding surge-type glaciers in the XM.

Topographic maps have been proven as suitable when assessing glacier areas and mass changes. The Shuttle Radar Topography Mission (SRTM) DEM and single-pass X-band InSAR from the TerraSAR-X and TanDEM-X digital elevation

measurements have yielded good results regarding glacier elevation change measurements. We used these data in our study to assess the glacier mass changes in the XM from 1970/71 to 2011/12. We also analysed glacier area changes from 1970/71 to 2018 as derived from glacier inventories and Landsat images. In addition, glacier surge and/or advance in this region was also considered.

## 2 Study area

Xinqingfeng (also called Buka Daban Peak, see Fig. 1) is located in the centre Hoh Xili region of the Kunlun Mountain. It is a small ice cap that developed on the planation surface of Kunlun Mountain. The highest elevation is 6860 m above sea level (a.s.l.). According to the second Chinese glacier inventory (CGI), Xinqingfeng contains 77 glaciers with a total area of 425.4 km$^2$. The glaciers are distributed around the ice cap with short tongues and with an average terminal altitude of approximately 5,056 m. The largest glacier is located on the southeastern slope called Monuomaha Glacier (or Xinqingfeng Glacier, Glacier No.4), with an area of 83.9 km$^2$, and the second largest glacier is located on the northwestern slope called the West Monuomaha Glacier (Glacier No.2), with an area of 69.0 km$^2$.

The Malan Ice Caps (Fig. 1) are located on the southwest side of Xinqingfeng, where the highest elevation is 6,056 m a.s.l. According to the second CGI, the Malan Ice Caps contains 59 glaciers with a total of area of 189.7 km$^2$. The largest glacier (Glacier No.14) is located on the southern slope, with an area of 30.4 km$^2$.

The XM is weakly affected by the westerly circulation and monsoon circulation. Based on fieldwork of the Kumukuli basin on the north side of Xinqingfeng, the precipitation in the Kumukuli basin from June to July of 1985 was 216 mm, and annual precipitation was estimated to be above 300 mm. In this same report, the annual precipitation of the Taiyanghu basin on the side of the Malan Ice Caps was 170 mm, with approximately 80% of the precipitation concentrated during the warm season (from May to October) (Xie et al., 2000). Based on the first CGI, near the snowline of Xinqingfeng, the average elevation was 5,440 m, the annual average air temperature was -15.4 ℃, and the annual precipitation was 340 mm. The average elevation of the snowline at the Malan Ice Caps was 5445 m and the air temperature was estimated as -11.5 ℃.

## 3 Data and methods

### 3.1 Topographic maps

Four topographic maps with a 1:100,000 scale (sheet numbers: I-46-2, I-46-3, J-46-134 and J-46-135) were constructed from aerial photos taken between 1970/71 by the Chinese Military Geodetic Service (CMGS) and used in glacier outlines digitized for the first CGI. The contour lines of the three topographic maps, with the exception of J-46-134, were obtained from the CMGS and were georeferenced into the WGS84/EGM96 using a seven-parameter transformation method. These

were then interpolated into DEMs (hereafter, referred to as TOPO DEMs) with a spatial resolution of 30 m. These TOPO DEMs covered the entire Malan and a small portion of Xinqingfeng.

### 3.2 SRTM and C-band Radar penetration

The SRTM DEM were acquired from interferometry of C-band and X-band radar from the 11 to 22 of February 2000. These data can often be seen as representative of the glacier surface at the end of the 1999 ablation period with slight seasonal variances (Gardelle et al., 2013). The 1″ C band DEM (SRTM 1) and the 3″ C-band DEM (SRTM 3) are freely available and cover most of the globe. The X-band SAR system has a narrower swath width than the C-band SAR, and, unfortunately, we could not access any X-band SRTM DEM data in our study area. Therefore, we used the SRTM C-band DEM at EGM96 orthometric heights with a 30 m pixel resolution (SRTM 1) in our study. However, the penetration of the C-band radar into snow and ice needs to be considered when assessing changes in the glacier elevation using the SRTM C-band DEM (Gardelle et al., 2012a). The X-band penetration depth is generally negligible, and Zhou et al. (2019) assumed that the impact of the X-band penetration depth is not sufficient to subvert the geodetic mass balance in our study region. Thus, we estimated the C-band penetration by comparing the SRTM C-band with the SRTM X-band DEM (cf. Gardelle et al., 2012a) in two regions near our study region (northeast by 45 km and southeast by 25 km) where the glaciers are at an elevation of 4930-5950 m a.s.l.. Nearly 92% of the glacial region in the XM are in this elevation range. We also estimated the penetration for higher than 5,950 m a.s.l. or lower than 4,930 m a.s.l. using the observed linear trend (Fig. S1).

### 3.3 TerraSAR-X/TanDEM-X

The TerraSAR-X was launched in June 2007 followed by its twin satellite, the TanDEM-X, in June 2010. The two satellites fly in a close orbital formation to act as a flexible baseline configuration (Krieger et al., 2007). Five pairs of TerraSAR-X/TanDEM-X (TSX/TDX) data in the experimental Co-registered Single look Slant range Complex (CoSSC) format acquired in the bistatic InSAR stripmap mode were employed in our study (Table 1). The CoSSC product was focused and co-registered at the TanDEM-X Processing and Archiving Facility (PAF) using the integrated TanDEM processor. The GAMMA SAR and interferometric processing software were used to process the CoSS product (Neckel et al., 2013). In our study, the DEM from the TSX/TDX (TSX/TDX DEM) interferogram was calculated, and changes to the glacier elevation between the TSX/TDX and SRTM were determined using differential SAR interferometry (D-InSAR). For the bistatic mode, neither the deformation nor the atmospheric delay phase was included in the interferogram, and the phase resulting from noise was also be ignored.

The SRTM 1 and TSX/TDX were co-registered before constructing the differential interferogram. This required establishing an initial look-up table based on the relationship between the map coordinates of the SRTM 1 DEM and the SAR geometry of the TSX/TDX master file. The offsets between the master image and the simulated intensity image of the SRTM 1 DEM




used an optimization of the simulated SAR images by employing GAMMA's *offset_pwrm* module. The SRTM 1 DEM was then transformed into a SAR geometry for the TSX/TDX master image. The simulated interferometric phase ($\Delta\emptyset_{TSX/TDX}$) from the SRTM 1 DEM ($\Delta\emptyset_{SRTM}$) was subtracted from the interferometric phase of the TSX/TDX data. The D-InSAR phase ($\Delta\emptyset'_{diff}$) can be obtained from Eq. (1):

$$\Delta\emptyset'_{diff} = \Delta\emptyset_{TSX/TDX} - \Delta\emptyset_{SRTM} \,, \tag{1}$$

The differential interferogram was filtered using an adaptive filtering approach. The flattened differential interferogram was unwrapped using the Minimum Cost Flow (MCF) algorithm (Costantini, 1998). The areas of layover and shadow with low coherence (<0.3) were masked out during the unwrapping processing. The unwrapped differential phases were converted to absolute differential heights using the calculated phase-to-height sensitivity. The differential interference uncertainty caused

by baseline errors can be regarded as a systematic error. A two-dimensional first-order polynomial fit for the non-glacial regions was used to remove the residuals in the glacial regions. Finally, a map with a spatial resolution of 12 m from the SAR coordinates was geocoded to geographic coordinates using a refined look-up table.

### 3.4 ASTER

The ASTER sensor onboard the TERRA satellite platform provides a stereo pair generated by nadir-looking (3N, 0.76–0.86

μm) and backward-looking (3B, 27.7 ° off-nadir) cameras with a base-to-height (B/H) ratio of approximately 0.6. This value is close to ideal for generating DEMs with a variety of terrain conditions via automated techniques (Kamp et al., 2003). In this study, the ASTER DEMs (Table 1), which were generated using the DEM Extraction Model from the ENVI 5.0 software, were used in the glacier surge analysis.

### 3.5 Global Land Ice Velocity Extraction from Landsat 8 (GoLIVE) data

The Global Land Ice Velocity Extraction from Landsat 8 (GoLIVE) data set is a compilation of ice velocities derived from the cross-correlation of pixel positions in pairs of panchromatic Landsat 8 images acquired from May 2013 to the present (Fahnestock et al., 2016). We considered only the velocities of the peak correlation values (corr) >0.4 and the differences in correlation values between the primary and secondary peaks (del_corr) <0.3 (Sam et al., 2018). We calculated the mean or maximum velocities for all the velocity rasters over each of the different years. We then discarded any annual average

velocity pixels that were over 1 standard deviation from the mean velocity values.

### 3.6 Glacier boundary mapping, changing and uncertainty

The glacier boundaries from 1970/71 were derived from the first CGI, which was inventoried using topographic maps and verified using aerial and Landsat MSS images. The glacier boundaries from  2000, 2013 and 2018 were digitised manually



from Landsat images using the same method as the second CGI processing (Guo et al., 2015). We also checked the glacier

boundaries by cross-checking with Google Earth imagery.

The uncertainty in determining glacier boundaries ($E_a$) was estimated using a buffer of 13.5 m for the topographic maps with

a 1:100 000 scale and half a pixel for the Landsat images (Wei et al., 2014). The uncertainty of the glacier area change ($E_{ac}$)

was calculated using Eq. (2):

$$E_{ac} = \sqrt{E_{a1}^2 + E_{a2}^2} \, , \tag{2}$$

where $E_{a1}$ and $E_{a2}$ represent the uncertainties of the glacier areas for the two different periods.

### 3.7 Glacier length

In this study, we estimated the glacier lengths by generating glacier centrelines using an automated method. This approach

was based on the glacier axis as derived from the glacier morphology, which requires glacier outlines and DEMs as the

inputs (Yao et al., 2015). The glacier centreline was derived from the SRTM DEM and glacier outline with the largest area.

We split the glacier centreline with the glacier outlines in the different periods and calculated the associated glacier lengths.

Similar to the uncertainty of glacier boundaries, the uncertainty of glacier lengths was also estimated using a buffer of 13.5

m for the topographic maps with a 1:100,000 scale and half a pixel for the Landsat images. The final uncertainty was also

calculated with Eq. (2), where $E_{ac}$ represents the uncertainty in the changes of the glacier lengths and $E_{a1}$ and $E_{a2}$ represent

the uncertainties of the glacier lengths from two different times.

### 3.8 Glacier elevation changes, mass balance and uncertainty

Changes in glacier elevation from 1999 to 2011/12 were calculated using the D-InSAR based on the TSX/TDX and SRTM

C-band (see Section 3.3). Changes in the glacier elevation from 1970/71 to 1999 and from 1970/71 to 2011/12 were

calculated by taking the differences between the DEMs (Nuth and Kääb, 2011) for the TOPO DEM, SRTM and TSX/TDX

DEM. Before the DEM differencing, the DEMs were corrected for planimetric and altimetric shifts using the TOPO DEM as

a reference. Then, the curvature bias (Gardelle et al., 2012a) in the glacial region was corrected by fitting sixth-order

polynomials to the elevation differences for the non-glacial regions. We considered only the elevation differences between

±100 m over the stable region with slopes ranging from 5° and 75° and excluding glaciers and water bodies in the co-

registered and bias corrected DEMs. Changes in the glacier mass were calculated based on variations in the glacier surface

elevation using an assumption or model for the density. We approximated the density as 850±60 kg m$^{-3}$ as a reasonable and

widely used assumption over a longer time periods (Huss, 2013).

The uncertainty in the differences of the glacier elevation ($E_{\Delta H}$) was estimated using the mean elevation difference ($E_{med}$) and

the standard deviation ($\sigma$) of the stable region for each altitude band (50 m), which excluded glaciers and water bodies:

$$E_{\Delta H} = \sqrt{E_{med}^2 + \sigma^2/N_{eff}} \, , \tag{3}$$



$$N_{eff} = N_{tot} \cdot PS/2d ,\qquad (4)$$

where $N_{eff}$ is the effective number of observations and is calculated using the total number of observations ($N_{tot}$), the pixel

size (PS, 30 m) and d, the distance for the spatial autocorrelation of the elevation change maps (1410 m) was determined

using Moran's I autocorrelation index for the elevation differences of non-glacierized region (Bolch et al., 2011; Gardelle et

al., 2013). The overall uncertainty of the DEM difference is the average of $E_{\Delta H}$ as weighted by the glacier hypsometry.

The uncertainty of the glacier boundaries ($E_a$) should be considered in the mass balance estimation. The glacier outlines were

used as they have a larger extent in the investigated period. The uncertainty in the radar penetration ($E_p$) also should be

considered in the mass balance estimation. We used the uncertainty in the DEM difference between the SRTM-X and

SRTM-C using Formula (3) to represent $E_p$, and the results revealed an uncertainty of 1.9 m. Finally, the uncertainty of the

volume to mass conversion should also be considered to calculate the final uncertainty ($Em$ of $\pm 60$ km m$^{-3}$ for the elevation

change to mass change) (Bolch et al., 2017):

$$E = \sqrt{E_{\Delta H}{}^2 + \left(\frac{\Delta H \cdot E_a}{S}\right)^2 + E_p{}^2 + \left(\frac{\Delta H \cdot E_m}{\rho}\right)^2} ,\qquad (5)$$

where $S$ represents the glacier area and $\rho$ represent ice density.

## 4 Results

### 4.1 Glacier area and length changes

There were 136 glaciers in XM with a total area of 641.2±7.7 km$^2$ in 1970/71. The glaciers are primarily located in the

southern and northern slopes (Fig. 2), and nearly 89% of the glacier areas lay between 5,100–5,900 m a.s.l. (Fig. 3). The

maximum elevation of the glaciers at Xinqingfeng (6,821 m a.s.l.) is higher than at Malan (6013 m a.s.l.), and the mean

median elevation (5,552 m a.s.l. in 1970/71, 5,560 m a.s.l. in 2018) is likewise higher (5,525 m a.s.l. in 1970/71, and 5,533

m a.s.l. in 2018).

The total glacier area decreased by 27.4±8.9 km$^2$ (4.3±1.4%) or 0.09±0.03% a$^{-1}$ from 1970/71 to 2018. The glacier area only

decreased by 0.01±0.03% a$^{-1}$ from 2013 to 2018, but it decreased by 0.03±0.03% a$^{-1}$ and 0.13±0.03% a$^{-1}$ for the periods from

2000-2013 and 1970/71-2000, respectively. The glacier area at Xinqingfeng decreased by 0.08±0.03% a$^{-1}$, while at Malan it

decreased by 0.11±0.03% a$^{-1}$. The glaciers showed heterogeneous variations with some advanced or surged (Tables 2 and 3).

Overall, the shrinkage speed of the glacier area decreased after 2000 and the glacier area was stable after 2013, which can be

mainly attributed to the heterogeneous variations with some of the glaciers advancing or surging (Tables 2 and 3).

There were no observed changes that occurred for the glaciers located above 5,600 m a.s.l. (Fig. 3). The glaciers in the

western and northeastern slopes experienced the most shrinkage at Xinqingfeng (Fig. 4). However, our result showed there

was little change in the glacier area of the north-western slope as a result of the advancing West Monuomaha Glacier (No. 2).

We also found that glaciers located on the northern slope experienced less shrinkage (Fig. 4) as a result of the advancement


of Glacier No. 1. Although the Monuomaha Glacier (No. 4) facing the eastern slope advanced from 2000-2018, it also experienced the most area shrinkage (Tables 2 and 3). Both Glacier Nos. 6 and 7 facing the southern slope experienced area loss over the entire investigated period. Thus, glaciers facing eastern and southern slopes experienced more shrinkage than those facing the northern and western slopes (Fig. 4). The glaciers along the eastern slope experienced the most shrinkage at

Malan while the northeastern glaciers were ranked second (Fig. 4). It is noted that most of the glaciers are located on the northern and southern slopes at Malan (Fig. 2). However, glaciers facing the northern slope lost more area (Fig. 4).

**4.2 Glacier mass changes**

The average elevation decrease of glaciers in Xinqingfeng was -3.50 $\pm$ 2.17 m, resulting in an average glacier mass loss of -0.22 $\pm$ 0.17 m w.e. a$^{-1}$ from 1999-2011/12. The highest mass loss was observed for the northwestern, northern, northeastern

and eastern slopes where the mass budgets were in the range of -0.25 $\pm$ 0.17 to 0.33 $\pm$ 0.17 m w.e. a$^{-1}$. In particular, the glaciers with a north-western aspect showed a relatively strong mass loss at Xinqingfeng from 1999-2011/12 (Fig. 6). West Monuomaha Glacier, which contributes to ~63% of the ice cover in the north-western slope, began to surge or advance after 1998, and experienced a significant lowering on the tongue, resulting in a net mass loss from 1999-2011 (Fig. 5). It is noted that most of the glaciers are located on the northern, northwestern, eastern and southern slopes at Xinqingfeng (Fig. 2).

However, the glaciers facing the southern slope showed only a minimal mass loss. Among these glaciers, Glacier Nos. 5-7 had mass gains of 0.01 $\pm$ 0.16 to 0.10 $\pm$ 0.16 w.e. a$^{-1}$ (Fig. 5 and Table 4).

The glaciers at Malan decreased in elevation from 1970-2012 with an average of thinning of 10.72 $\pm$ 0.91 m, resulting in an average mass loss of 0.22 $\pm$ 0.02 m w.e. a$^{-1}$. The rate of mass loss for these glaciers increased from -0.19 $\pm$0.14 m w.e. a$^{-1}$ from 1970-1999 to -0.29 $\pm$0.17 m w.e. a$^{-1}$ from 1999-2012. The glaciers at different slopes showed similar mass losses from

1999-2011/12 (Fig. 6). However, glaciers facing the northern slope showed a greater mass loss budget than the southern slope from 1970-1999 and from 1970-2011/12 (Fig. 7). This is because Glacier Nos.14 and 15 at the southern slope experienced a slightly positive mass budget from 1970-1999.

**4.3 Glacier velocity from 2013-2018**

The average velocity for the glaciers in XM from 2013-2018 was 0.16 m d$^{-1}$, and the average velocity for glaciers at

Xinqingfeng (0.17 m d$^{-1}$) was higher than at Malan (0.14 m d$^{-1}$). This is because there is a higher average slope for the glaciers at Xinqingfeng than at Malan. There were nearly no changes in the glacier velocities except for Monuomaha Glacier and Zu Glacier (No. 6) from 2013-2018.

The Monuomaha Glacier experienced a larger velocity (0.8 m d$^{-1}$) from 2013-2016, and then returned back to its normal levels from 2017-2018 (Figs. 8a and 8b). The maximum velocity of Monuomaha Glacier was 1.8 m d$^{-1}$. We found that

Monuomaha Glacier showed significant thickening within its tongues from 1999-2018, resulting in an advancing with from

2010-2016, which is typical for a surge (Fig. 8c). However, its tongues also showed a significant thinning and melted completely, resulting in a retreating from 1971-1999 (Fig. 8d).

The Zu Glacier showed a slightly higher velocity from 2013-2015 (Figs. 9a and 9b), and there was a slight thickening in its lower parts from 2011-2014 (Fig. 9c). Moreover, a significant thickening in its tongue and thinning in its upper parts were

found from 2014-2018. We assert that this behaviour is indicative of a surge-type glacier. A slight thickening in the upper parts was also found from 1999-2011. Our results (Table 4) suggest that the Zu Glacier experienced a positive mass balance from 1999-2011. Thus, its surge initiation could have been produced from 1999-2012. Similarly, the glacier also surged before 1999 due to a thickening in the tongue and thinning in the upper parts. However, we could not determine the exact timing of this surge due to a lack of data. A significant thinning in its tongue from 1999-2011 confirmed that the active surge

event (1971-1999) transferred ice mass to lower elevations where it was more prone to melting during this period.

### 4.4 Glacier advance and surge

A minimum of seven glaciers at Xinqingfeng and Malan showed heterogeneous variations with either surging or advancing at different periods (Table 5). The eastern branch of Glacier No. 1 advanced 278.4 $\pm$ 21.2 m and converged into the West Monuomaha Glacier from 1971-1987 at the same time that the West Monuomaha Glacier was retreating. Glacier No. 1 then

continued to advance 50.9 $\pm$ 21.2 m from 1989-1999. The average advance rate for Glacier No. 1 from 1971-1999 was 11.8 m a$^{-1}$. The West Monuomaha Glacier then advanced 1,200 $\pm$ 21.2 m from 1987-1989 and continued to advanced 256 $\pm$ 21.2 m from 1990-1998. The average advance rate for the West Monuomaha Glacier for this entire period from 1987-1998 was 132.4 m a$^{-1}$, where the peak value was 600 m a$^{-1}$ from 1987-1989. The Monuomaha Glacier also advanced 1,164.0 $\pm$ 16.8 m from 2010-2016; however, it had an overall retreat of 2,546.8 $\pm$ 20.2 m over the entire period from 1970-2010. In fact, surge

initiation of Monuomaha Glacier began in 2009 while the ice at the north side decreased.

The glacier velocities from the Landsat dataset (Fig. S2) suggest that the Monuomaha Glacier could have been initiated at some point between February and March, 2009. We also found that the velocity of Monuomaha Glacier fell abruptly in January, 2017 and then likely returned to normal levels by March, 2017 based on the time series data for glacier velocities derived from GoLIVE. The Zu Glacier advanced 46.0 $\pm$ 10.6 m from 2014 to 2016. Glacier No. 7 advanced 40.8 $\pm$ 20.2 m

from 1986 to 1989, and then advanced suddenly by 107.8 $\pm$ 16.8 m from 2009-2010. Glacier No. 8 advanced 432.0 $\pm$ 20.2 m from 1970-2000; however, we only know for certain that this glacier advanced 663.0 $\pm$ 20.2 m from 1970-1986 and then retreated between 1986 and 2018 due to lack of data. Glacier No. 14 showed significant thickening and thinning within its tongues, which is typical of a surge, over the entire study period. We found that Glacier No. 14 advanced 260.5 $\pm$ 16.8 m from 2007-2012 but retreated 951.0 $\pm$ 20.2 m from 1971-2007.

The latest surge-type index from Mukherjee et al. (2018) would classify the West Monuomaha Glacier, Monuomaha Glacier and Glacier No.7 as surge-type glaciers (>100 m a-1). Our results showed that the Monuomaha Glacier and Glacier Nos. 7, 8



and 14 showed significant thickening in their tongues and a significant lowering in their upper parts (Fig. 5b). We confirmed that Glacier No. 14 surged as a consequence of its advance from 2007-2012. While Glacier No. 8 is considered a surge-type glacier with surging from 1999-2011, it is unclear whether its advance from 1970-1986 was caused by surging. In addition, we also found that Glacier No. 1 is likely a surge-type glacier (<100 m a$^{-1}$ and >10 m a$^{-1}$).

## 5 Discussion

### 5.1 Glacier area changes

Our results show a decrease in the glacier area at a rate of ~0.00-0.26% a$^{-1}$ (with the exception of Glacier Nos. 1,2 and 8) from 1971-2018. These results are in agreement with other studies, proving that there are low rates of glacier shrinkage in the inner Tibetan Plateau. For example, there was an 0.18% a$^{-1}$ shrinkage from the 1970s to 2009 for the drainage Basins of Ayakkum Lake (5Z11, basin code of which we take from CGI) and Hoh Xil Lake (5Z12), 0.17% a$^{-1}$ from 1976-2013 for the Qaidam interior-drainage basin (5Y5), and 0.14 % a$^{-1}$ from 1976-2013 for the Ayakkum lake interior-drainage basin (5Z1) (Wei et al., 2014; Ye et al., 2017).

Compared with the surrounding regions in the Tibetan Plateau, the rate of glacier shrinkage in XM was very close to the western Kunlun Shan (0.1% a$^{-1}$ from 1970-2010) (Bao et al., 2015) and Kangzhag Ri (0.08% a$^{-1}$ from 1970-2016) to the west (Zhang and Liu, 2018) was slightly lower than Geladandong (0.15% a$^{-1}$ in the 1964-2010) (Wang et al., 2013) to the south, and was significantly lower than Dongkemadi (0.26% a$^{-1}$ from 2000-2011) (Qiao, 2010) and Qilian Shan (0.39% a$^{-1}$ from 1956-2010) (Sun et al., 2018) to the northeast, Bugyai Kangri (0.48% a$^{-1}$ from 1981-2013) (Liu et al., 2015) to the southeast and western Nyainqentanglha (0.62% a$^{-1}$ from 1970-2014) (Wu et al., 2016) to the south. Reductions to the glacier area showed a trend of low to high from the western to eastern and are consistent with the isothermal line trend (Xie et al., 2000), indicating that XM maybe a turning point in this trend.

### 5.2 Glacier mass changes

Our results for the mass changes of the glaciers at Xinqingfeng and Malan of −0.22 ± 0.17 m w.e. a$^{-1}$ and −0.29 ± 0.17 m w.e. a$^{-1}$, respectively, from 1999-2011 agree well with the results from Zhou et al. (2019) of −0.21 ± 0.10 m w.e. a$^{-1}$ and −0.22 ± 0.10 m w.e. a$^{-1}$ from 2000-2016. Brun et al. (2017) reported that the global average mass loss was 0.14 ± 0.07 m w.e. a$^{-1}$ from 2000-2016 for glaciers in the inner TP. We also estimated that there was a mass loss of 0.17 m w.e. a$^{-1}$ for the glaciers at XM based on the data from Brun et al. (2017) (Table 5), which agrees well with our study. However, Gardner et al. (2013) also found a −0.01 ± 0.35 m a$^{-1}$ elevation change for the glaciers of the inner Tibetan Plateau from 2003-2009 using ICESat and SRTM. This deviation may be attributed to the different study periods and the uncertain penetration of the SRTM C-band radar into the ice and snow. Neckel et al. (2014) determined that the average mass loss was 0.77 ± 0.35 m w.e. a$^{-1}$ from

2003-2009 for glaciers in the Qilian Mountains and East Kunlun, which are included in our study region, as observed by ICESat GLAS. Their estimated trend is more significant than ours, which may be attributed to a different study extent and time period. These results together prove that the very large inner Tibetan Plateau is an aggregation of climatically heterogeneous sub-regions that result in spatial variability in the glacier mass balance. Previous studies (Bao et al., 2015; Lin

et al., 2017) have reported that the west Kunlun Shan and extended West Kunlun show glacier mass gain from 2000. Zhang and Liu (2018) determined that glaciers in Kangzhag Ri, which are approximately 70 km west of XM, showed a positive mass balance (+ 0.16 ± 0.02 m w.e. a$^{-1}$) from 1999 to 2012 using ASTER and SRTM. This appears to suggest that KangzhagRi and XM are the transition zones from the west to the east regarding the mass balance distribution from positive to negative based on the data since 1999.

Most glaciers experienced similar mass budgets for the investigated periods at Malan. However, it seems that some glaciers had greater negative budgets after 1999, e.g., Malan experienced a slight mass gain before 1999 and a negative mass change after 1999. The global average mass change trend (more negative) is in agreement with Bugyai Kangri, Dongkemadi and West Geladandong to the southeast of XM (Chen et al., 2017). However, this is in contrast to KangzhagRi (Zhang and Liu, 2018), Aru Co to (Zhang et al., 2018) to the west of XM, which showed more positive growth after a negative period. This

could be explained if XM was a turning point in the observed trends. Glaciers to the west of XM showed less mass loss or a mass gain trend after 1999, but glaciers to the east showed a more negative change.

**5.3 Glacier advance and surge**

Elevation profiles (Fig. 10) indicate that most of the surged or advanced glaciers have greater slopes over the accumulation area and gentler slopes over the ablation area. This is in accordance with the topographic features of surged glaciers that

exhibit small slopes with velocities that are too low to remain in balance with the accumulation rate (Björnsson et al., 2003). Thus, surge is necessary to transport mass from the reservoir area down the glacier tongue to the terminus. Glacier Nos.6-8 were nearly balanced from 1999-2012. These glaciers have relatively narrow width, so the mass transported from the reservoir area might be limited. As a result, these glaciers experience only small advances. Although the mass transported from higher to lower elevations was more vulnerable to increasing temperatures, the observed small ice mass losses have not

been enough to result in a strong negative mass balance. However, others glacier with wider accumulation areas would provide more ice mass to the terminus, which then results in a more negative mass balance. In contrast to the other glaciers, the Monuomaha Glacier has a gentle accumulation area and a greater slope over the ablation area. Similarly, a gentle accumulation area results in smaller velocities to retain a greater ice mass with the accumulation, which prohibits the ice mass from transferring to the ablation area. Therefore, the terminus was illsupplied from the accumulation area and exhibited

a sharp retreat from 1970-2010. Once the glacier surged, a significant amount of ice mass was transferred from the accumulation area to the terminus and the glacier advanced more because of the greater slope. However, the ice over the



terminus with a greater slope and lower elevation could result in thinning and would be more vulnerable to increasing temperatures. Hence, the Monuomaha Glacier showed a retreated for most of 1970 to 2010, but it could have surged before 1970. We also predicted that the Monuomaha Glacier will retreat sharply again in the future.

Two surge models have been linked to a corresponding hypothesis relative to thermal (Svalbard-type) or hydrological (Alaska-type) surge control (Falaschi et al., 2018; Quincey et al., 2015). Thermal control is characterized by an initiation phase that lasts several years before reaching a peak in the surge and a termination phase that consists of several years of deceleration following the surge peak (Clarke et al., 1984; Murray et al., 2000). These surges can begin or end at any seasonal time of year. Hydrological control is characterized by the rapid acceleration and deceleration over a short time (i.e., days to weeks long) and tends to initiate during the winter months and terminate during the summer months (Burgess et al., 2012; Lingle and Fatland, 2003). Our results showed that the Monuomaha Glacier had a long duration active phase that lasted 8 years and may have begun and ended in the winter. We also estimated the contributions of the internal ice deformation $u_d$ to the surface flow with a parallel-sided slab assumption with the plain strain approximation, as

$$u_d = \frac{2A}{n+1}(\rho g sin\alpha)^n H^{n+1} , \qquad (6)$$

where A is strain rate factor at $2.4 \times 10^{-24}$ s$^{-1}$Pa$^{-3}$ (a conservative estimate), $\rho$ is the ice density at 900 kg m$^{-3}$, n is Glen's exponent at 3, g is gravitational acceleration at 9.8 m s$^{-2}$, $\alpha$ is the slope, and H is the ice thickness (Round et al., 2017).

A constant glacier thickness of 150-190 m was assumed based on the volume estimation from the second CGI. Our results show a 6.5 ° surface slope over the glacier tongue in 2000, which would result in deformation velocities of around 0.05-0.13 m d$^{-1}$. This estimate was close to the mean velocity of 0.09 m d$^{-1}$ from Jun 15, 2008 to Feb 10, 2009 as observed from remote sensing data (Fig. S3). The glacier surged from Feb, 2009 with a mean glacier velocity over the glacier tongue reaching 0.9 m d$^{-1}$ from Feb 10, 2009 to Mar 14, 2009 (Fig. S2) and increased to 1.2 m d$^{-1}$ in April (Fig. S4) with a peak velocity of 3.8 m d$^{-1}$ in November (Fig. S5). The glacier tongue velocity then reduced to 0.4 m d$^{-1}$ from 17 March to 4 May, 2010 (Fig. S6). The mean slope of the glacier tongue increased to 12.3 ° on 29, November, 2010, resulting in a deformation velocity of 0.9 m d$^{-1}$, which was close to the mean velocity observed from 17 March to 4 May, 2010. The velocities of the glacier tongue from 2013 to 2016 (Fig. 8) seem to be the same order of magnitude as the velocities through the internal deformation alone. Hence, it appears that the internal ice formation contributed significantly to the glacier tongue flow. This indicate that the Monuomaha Glacier might be surge controlled via thermal mechanisms, where a switch from cold to temperate conditions may have caused the surge onset in 2009.

The West Monuomaha Glacier surged from 1987-1989 and advanced from 1990-1998, but no advance or surge was found after 1999. Two advances occurred over a short time interval, which we assumed were in the same surge period and that the first period was during the peak of the surge. Our results show that the West Monuomaha Glacier retreated from 1971-1986. Liu et al. (2004) also reported that this glacier retreated from 1971-1976 in response to the higher air temperatures from

1934-1976. Although our results show that Glacier No. 1, which is adjacent to the West Monuomaha Glacier, experienced advancing from 1971-1986, it actually may have advanced from 1977-1987 after retreated from 1971-1976. Similarly, Glacier No. 1 advanced from 1989-1999. Thus, we assumed that Glacier No. 1 also surged from 1977 to 1987 and then decelerated until 1999, which is similar to the behaviour of the West Monuomaha Glacier. These glaciers also fit the

characteristics of thermal control, which began and ended over a long duration.

Glacier No. 7 was observed to advance from 1986-1989 and from 2009-2010, the latter of which we confirmed as being controlled by surging. If we assume that the first was also controlled by surging, the cycle (the active and quiescent phases) is only 10 years. However, as mention above, the cycles of Glacier No. 1, the West Monuomaha Glacier and the Monuomaha Glacier seem to be over a long time with a much longer active phase. Thus, it is hardly possible to precisely

determine the timing and duration of surge cycles due to deficiencies in the existing data. For each surging glacier, the active and quiescent phases tend to be of relatively constant length, resulting in a quasi-periodic cycle, although there are large variations in the cycle lengths between glaciers and between regions (Björnsson et al., 2003). For example, in Svalbard, which is surge controlled via thermal mechanisms, the active phase of surging glaciers typically lasts for 4–10 years, compared with only 1–3 years for surging glaciers in north-western Alaska, Iceland and the Pamirs. The maximum ice

velocities of the glaciers in Svalbard are comparatively low, ranging between 1.3 and 16 m d$^{-1}$, compared with velocities of 50 m d$^{-1}$ as measured for the Variegated Glacier in Alaska (Benn and Evans, 2010). Our results also show that the longer active and maximum ice velocity of the Monuomaha Glacier was similar to those at Svalbard. The quiescent phase is also relatively long for the Svalbard glaciers (50–500 years) compared with other areas (20–40 years). The length of the surge cycle for some glaciers is shown to reflect the time required for snow accumulation to refill the reservoir zone (e.g. Eisen et

al. (2001)). Therefore, the surge cycle will be shorter where snowfall rates are higher (such as in Alaska) compared with more arid regions (such as Svalbard). In this case, Glacier No. 7, which advanced from 1986-1989, may not be triggered from a surge mechanism. Instead, it could be a response to the colder and wetter climate that was seen from the late 1980s and into the 1990s (Wang, 2009; Wang et al., 2003). However, glacier surge in Pamir and Karakoram have variable controlling processes depending on the thermal and hydrological conditions and the geomorphological characteristics of

different individual glaciers (Lv et al., 2019; Quincey et al., 2015). It is possible that a similar glacier surge heterogeneity is applicable to the XM.

### 5.4 Glacier response to climate change

From the coldest years of the Little Ice Age, the areas of the glaciers at Malan were larger by 4.6% than for modern glaciers, compared with approximately 8% and 20% in Qangtang and TP, respectively (Pu et al., 2001). Thus, it is possible that the

glaciers in XM are more stable. However, the warming trend from the 20th century was recorded from a malan ice core, and the warmest period was during the 1950s to the early 1980s. There were also several stable cold periods that punctuated

through the warming, especially during the late 1980s to 1990s, which may have been caused by a strong summer monsoon (Wang et al., 2003). We could conclude that the shrinkage of glacier areas mainly occured from 1970-1999 and could be a response to the warming of the 1950s to the early 1980s. In addition, a higher net accumulation rate was recorded from 1987-1995 from an ice core in Malan (Wang, 2009). Thus, the colder and wetter climate from the late 1980s to 1990s could

result in a slight negative and even positive mass change for the glaciers during this period. This is likely the reason for the relatively small negative mass loss for the period of 1970-1999.

Glacier area shrinkage (1970-1999) may lag behind the mass loss from the 1950s to the early 1980s in response to a warmer climate. With the rapid warming that has been seen during the 21st century, glacier mass loss could be further accelerated. At the same time, glaciers have been observed to experience a slight area shrinkage from 2000 to 2018 relative to the colder

climate from the late 1980s and into the 1990s. There was also low precipitation in XM, and slight increases in precipitation more recently have had very little impact on glacier change in the 21st century.

**6 Conclusions**

We investigated Glacier area and mass changes in XM as derived from topographic maps, Landsat, ASTER, SRTM DEM, and TerraSAR-X/TanDEM-X for the period of ~1970-2018 and ~1970-2012, respectively. Our results show that the glaciers

experienced a small shrinkage from $641.2 \pm 7.7$ km$^2$ in 1970/71 to $613.9 \pm 4.4$ km$^2$ in 2018, corresponding to an area shrinkage of $4.3 \pm 1.4\%$ ($0.09 \pm 0.03\%$ a$^{-1}$) from 1970 to 2018. The shrinkage speed of the glacier area decreased after 2000 and the glacier area was stable after 2013, which can be mainly attributed to the advance or surge of some glaciers. However, the mass balances of glaciers at Xinqingfeng and Malan were negative at $-0.22 \pm 0.17$ m w.e. a$^{-1}$ and $-0.29 \pm 0.17$ m w.e. a$^{-1}$ from 1999-2012, respectively. A lower mass loss of $0.19 \pm 0.14$ m w.e. a$^{-1}$ was found for the glaciers at Malan from 1970-

1999. Glacier variations at XM are heterogeneous and differ spatially as well as temporally. Glaciers facing southern slopes showed only slight mass losses at Xinqingfeng, due to three main glaciers (Glacier No. 5, Zu Glacier, Glacier No. 7) showing a mass gain of $0.01 \pm 0.16 \sim 0.10 \pm 0.16$ w.e. a$^{-1}$. Glaciers at different aspects showed similar mass losses from 1999-2011/12; however, glaciers facing the northern slope experienced a more negative mass budget than at the southern slope, as two glaciers (Glacier Nos. 14 and 15) showed a positive mass budget from 1970-1999. A total of seven glaciers

showed surging or advancing from 1970-2018. Among them, the Monuomaha Glacier was active from 2009-2016 with a maximum velocity of 1.8 m d$^{-1}$ from 2013-2018, as compared with other glaciers with an average velocity of 0.16 m d$^{-1}$. These surge-type glaciers showed a long active period and comparatively low velocity, suggesting that thermal control is important for surge initiation and recession. The ablation area or accumulation area exhibited small slopes with velocities that were too slow to remain in balance with the accumulation rate and require surging to transport mass from the reservoir

area down to the glacier tongue.

*Author contributions.* The concept of this study was developed by Zhen Zhang and Shiyin Liu. The digital elevation models were generated by Zhen Zhang and Zongli Jiang. Zhen Zhang performed the data analysis and wrote the draft of the paper. Zhen Zhang, Shiyin Liu and all other authors were involved in paper writing or supported this work.

*Competing interests.* The authors declare that they have no competing interests.

*Acknowledgements.* This research was supported by the Strategic Priority Research Program of the Chinese Academy of Sciences (Grant No. XDA19070501), The Ministry of Science and Technology (Grant No.2013FY111400), International

Partnership Program of Chinese Academy of Sciences (Grant No. 131C11KYSB20160061), the National Natural Science Foundation of China (Grant Nos. 41701087, 41471067) and Research Funds Provided to New Recruitments of Yunnan University (YJRC3201702). Landsat, SRTM C-band and ASTER data were acquired from the US Geological Survey and NASA. The first and second glacier inventories were provided by a past MOST project (2006FY110200) (http://westdc.westgis.ac.cn/glacier). We thank DLR for access to SRTM X-band and TerraSAR-X/TanDEM-X data. We

also thank Etienne Berthier for guidance on uncertainty estimation of glacier elevation changes.

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





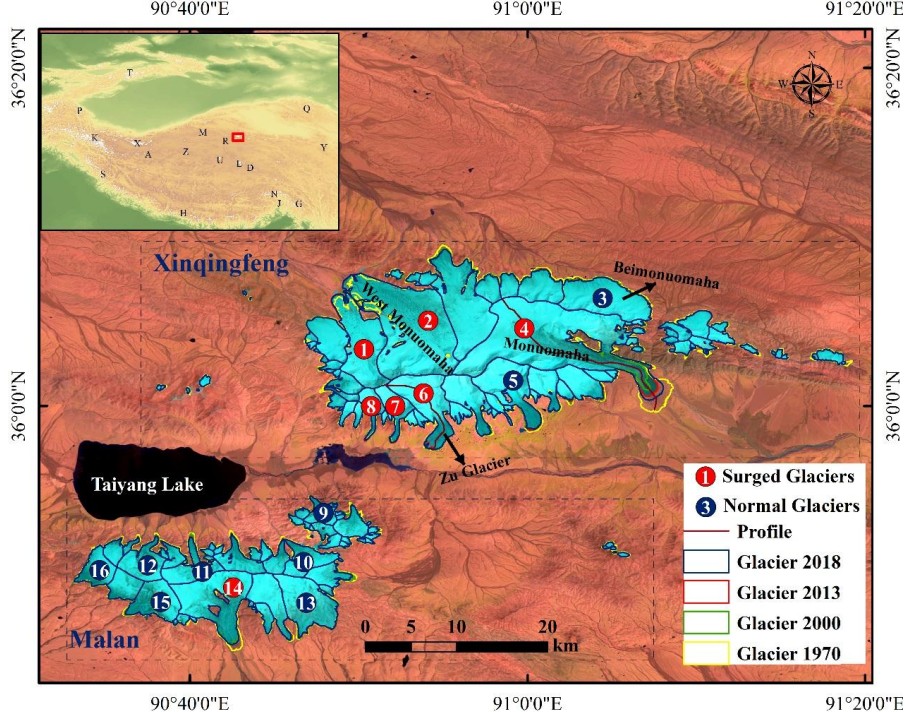

Fig. 1: Overview of the Xinqingfeng and Malan glaciers (Background image: Landsat 8 OLI of 31 July 2013, A: Aru Co, D: Dongkemadi, G: Kangri Karpo, H: Himalayas, J: Namjagbarwa, K: Karakoram, L: Geladandong, M:Ulugh Muztagh, N: Nyainqentangglha, P: Pamir, Q: Qilian, R: Kangzhag Ri, S: Spiti Lahaul, T: Tien Shan, U: Purogangri, X: West Kunlun, Y:

5    Ány êmaq ên, Z: Zangsar Kangri).





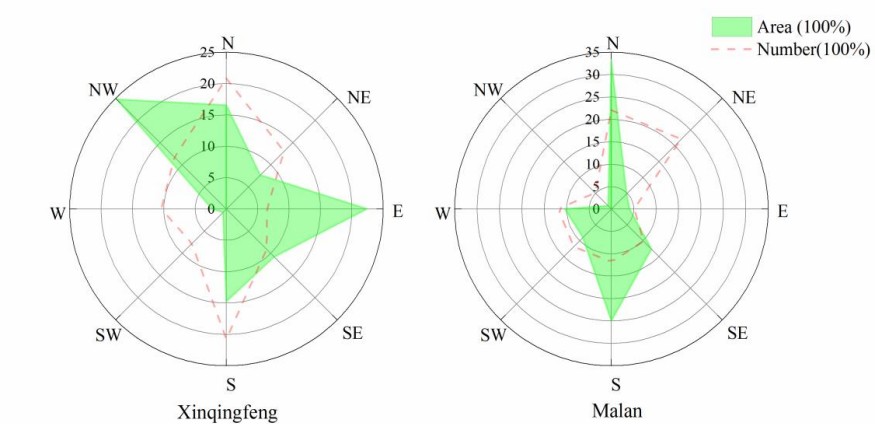

Fig. 2: Diagram showing the number and area covered for different aspect of the glaciers.





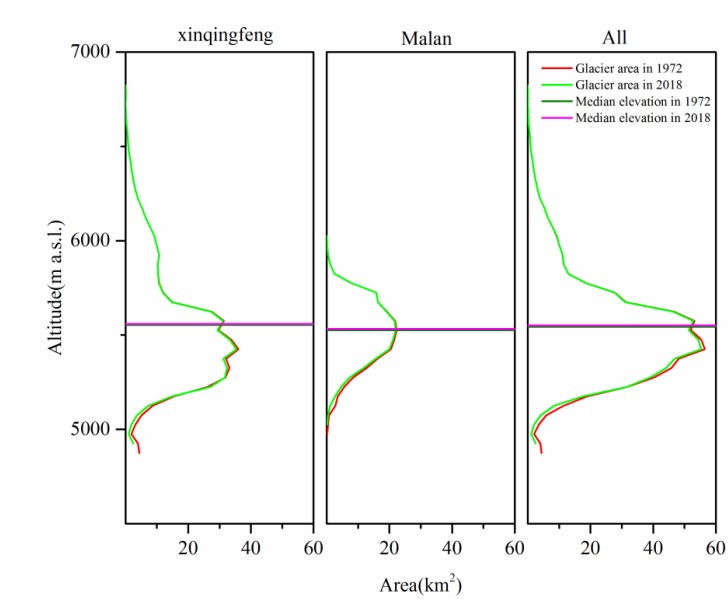

Fig. 3: Hypsography of glaciers in 1970/71 and 2018.





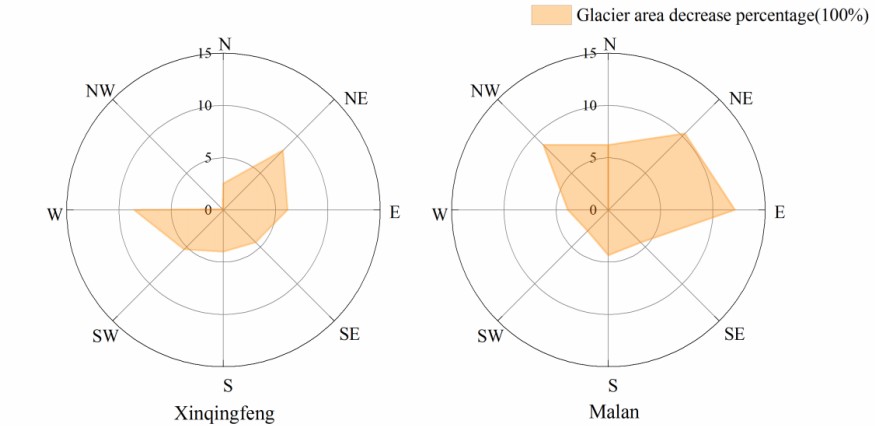

Fig. 4: Glacier area decreases in different aspects at Xinqingfeng (Left) and Malan (Right).





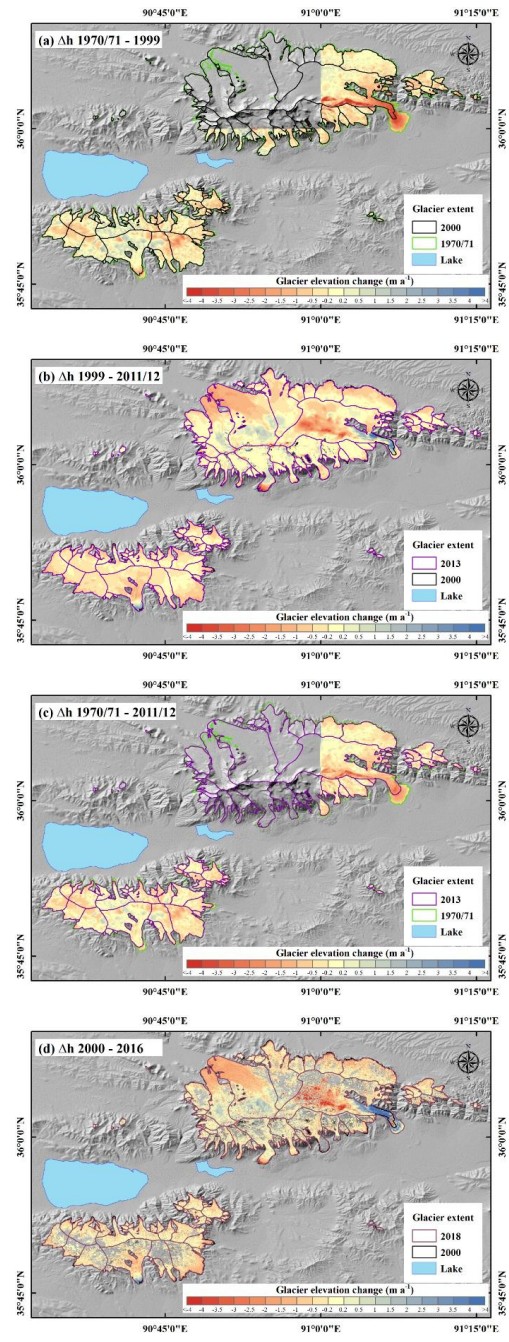

5   **Fig. 5: Elevation change of glaciers in XM during 1970/71-1999 (a), 1999-2011/12 (b), 1970/71-2011/12 (c) and 2000-2016 (d). The data of Fig. 5 (d) was derived from Brun et al. (2017).**



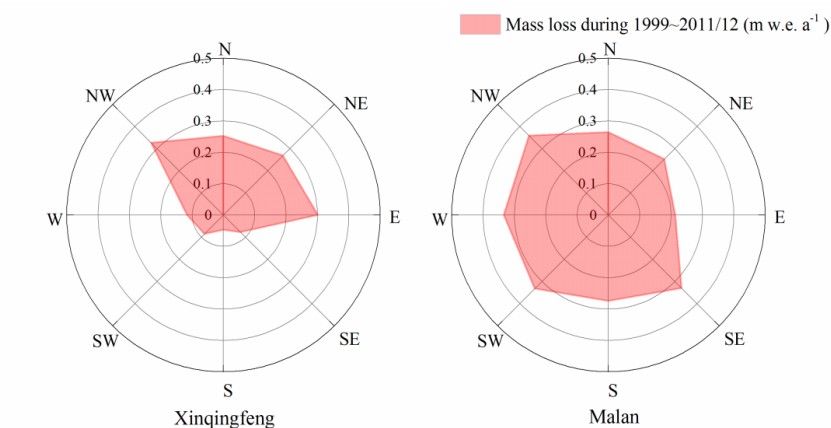

Fig. 6: Glacier mass loss in different aspects at Xinqingfeng (Left) and Malan (Right) during 1999-2011/12.





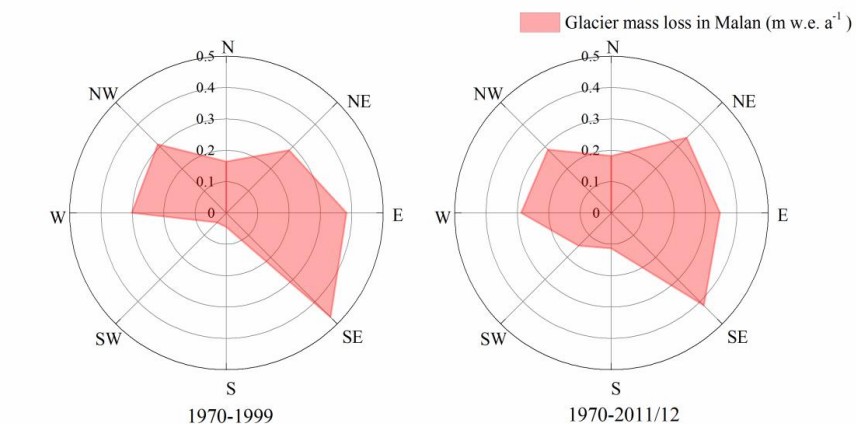

Fig. 7: Glacier mass loss in different aspects at Malan during 1970-1999 (Left) and 1970-2011/12 (Right).



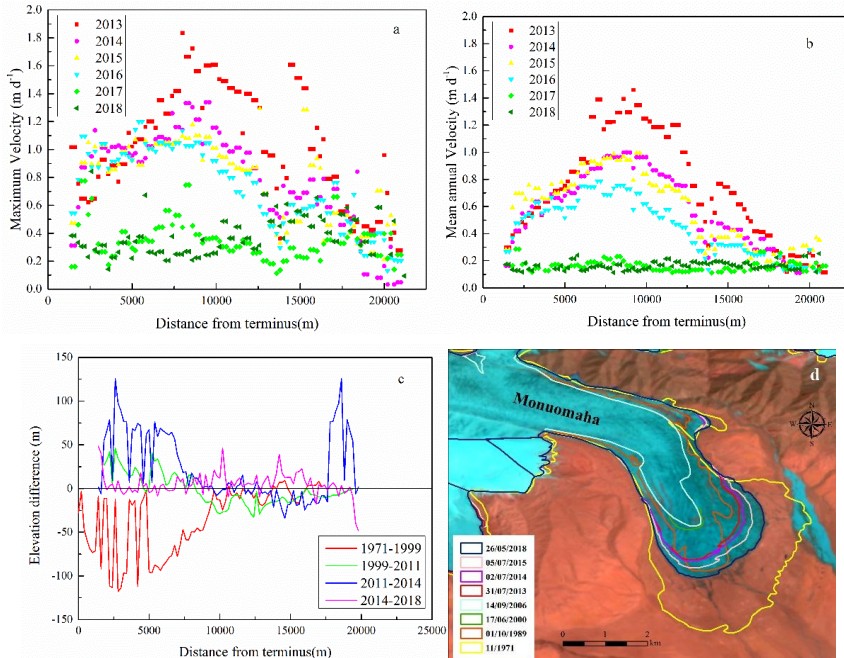

**Fig. 8: Panels (a)–(c) show Maximum (8a) and mean (8b) surface velocity and elevation difference (8c) profiles of Monuomaha Glacier during different periods. The profiles, derived from TOPO DEM data, follow the longitudinal path from Fig. 1. (d) shows**

5  **glacier terminus change. Background image from Landsat 8 OLI of 26 May 2018.**



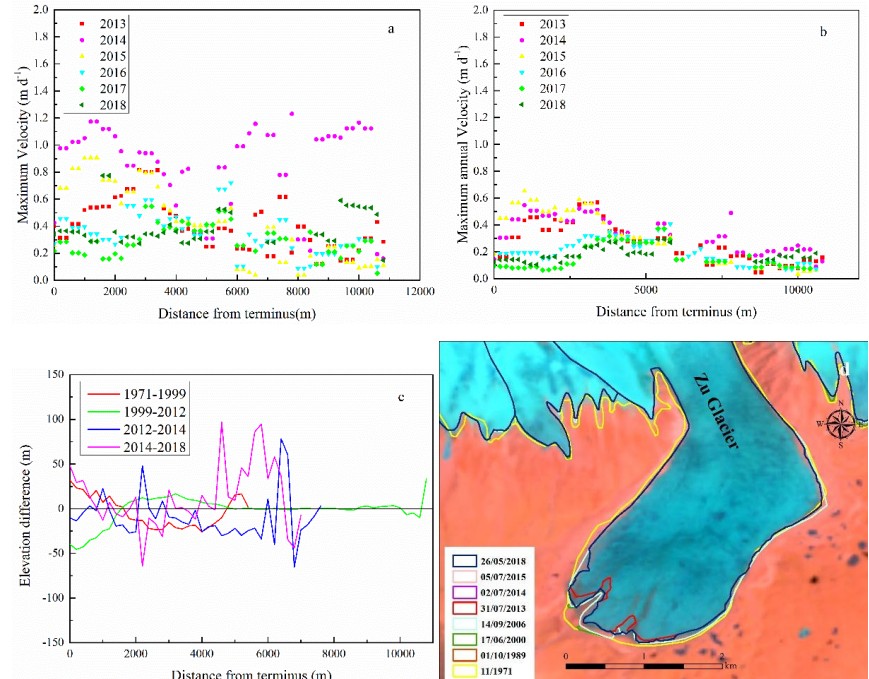

**Fig. 9: Panels (a)–(c) show Maximum (8a) and mean (8b) surface velocity and elevation difference (8c) profiles of Zu Glacier during different periods. The profiles, derived from TOPO DEM data, follow the longitudinal path from Fig. 1. (d) shows glacier terminus change. Background image from Landsat 8 OLI of 26 May 2018.**





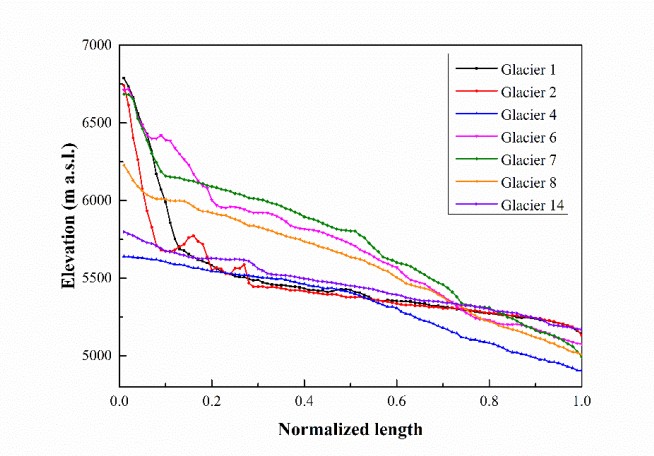

**Fig. 10: Elevation profiles of surged or advanced glaciers extracted from SRTM DEM. The glacier length is derived from glacier inventory of 2000.**



**Table 1.** Detailed information about the data used in this study

| Source | Acquisition date | Space Resolution (m) | Usage |
|---|---|---|---|
| TerraSAR-X/TanDEM-X | 8 Mar 2011 | 12 | Estimation of glacier |
| | 5 Mar 2012 | | elevation change |
| | 16 Mar 2012 | | |
| | 29 Apr 2012 | | |
| Topographic maps | Oct 1970 | | Glacier identification |
| | Jan 1971 | | |
| | Nov 1971 | | |
| | Dec 1971 | | |
| TOPO DEMs | Oct 1970 | 15 | Estimation of glacier |
| | Nov 1971 | | elevation change |
| | Dec 1971 | | |
| ASTER | 24 Jan 2014 | 15 | Estimation of glacier |
| | 25 Apr 2018 | | elevation change |
| SRTM DEM | 11-22 Feb 2000 | 30 | Estimation of glacier |
| (C-band and X-band) | | | elevation change |
| Landsat 1~3/MSS | 1972~1976 | 79 | Glacier identification |
| Landsat 5/TM | 1986~2011 | 30 | Glacier identification |
| Landsat 7/ETM+ | 2000~2012 | Pan: 15; MS: 30 | Glacier identification |
| Landsat 8/OLI | 2013~2018 | Pan: 15; MS: 30 | Glacier identification |
| GoLIVE | 2013~2018 | 300 | Glacier velocity |





**Table 2.** Glacier area (A) and changes (ΔA) from 1970−2018 for selected glaciers that have mass−balance estimates and for all glaciers of the study area.

| Region | ID | GLIMS ID | $A_{1970}$ (km²) | 1970−2000 | | | 2000−2013 | | | 1970−2013 | | | 2013−2018 | | | 1970−2018 | | |
|---|---|---|---|---|---|---|---|---|---|---|---|---|---|---|---|---|---|---|
| | | | | ΔA (km²) | ΔA (%) | Rate (% a⁻¹) | ΔA (km²) | ΔA (%) | Rate (% a⁻¹) | ΔA (km²) | ΔA (%) | Rate (% a⁻¹) | ΔA (km²) | ΔA (%) | Rate (% a⁻¹) | ΔA (km²) | ΔA (%) | Rate (% a⁻¹) |
| Xinqingfeng | 1 | G090837E36060N | 26.7 | 1.9 | 7.2 | 0.23 | −0.4 | −1.4 | −0.11 | 1.5 | 5.5 | 0.13 | −0.2 | −0.6 | −0.11 | 1.4 | 4.9 | 0.11 |
| | 2 | G090884E36076N | 66.6 | 2.4 | 3.6 | 0.12 | −0.4 | −0.6 | −0.05 | 2.0 | 2.9 | 0.07 | −0.0 | −0.0 | −0.00 | 2.0 | 2.9 | 0.06 |
| | 3 | G091076E36106N | 27.2 | −1.0 | −3.5 | −0.12 | −0.5 | −1.9 | −0.15 | −1.5 | −5.6 | −0.13 | −0.0 | −0.2 | −0.03 | −1.5 | −5.8 | −0.12 |
| | 4 | G091032E36060N | 94.8 | −10.8 | −11.4 | −0.40 | 3.6 | 4.3 | 0.33 | −7.2 | −8.2 | −0.18 | 1.5 | 1.7 | 0.34 | −5.7 | −6.4 | −0.13 |
| | 5 | G090983E36018N | 22.2 | −0.6 | −2.6 | −0.09 | −0.2 | −1.1 | −0.09 | −0.8 | −3.9 | −0.09 | −0.2 | −0.8 | −0.17 | −1.0 | −4.7 | −0.10 |
| | 6 | G090901E36002N | 23.3 | −0.4 | −1.5 | −0.05 | −0.3 | −1.2 | −0.09 | −0.6 | −2.8 | −0.06 | 0.1 | 0.3 | 0.06 | −0.5 | −2.1 | −0.04 |
| | 7 | G090868E35998N | 8.5 | −0.2 | −2.4 | −0.08 | 0.1 | 1.1 | 0.08 | −0.1 | −1.4 | −0.03 | −0.1 | −1.0 | −0.19 | −0.2 | −2.4 | −0.05 |
| | 8 | G090846E36001N | 5.7 | 0.0 | 0.7 | 0.02 | 0.0 | 0.6 | 0.05 | 0.1 | 1.3 | 0.03 | −0.0 | −0.6 | −0.12 | 0.0 | 0.7 | 0.02 |
| subtotal | | | 443.0 | −16.9 | −3.8 | −0.13 | −0.2 | −0.0 | −0.00 | −17.1 | −3.9 | −0.09 | 0.5 | 0.1 | 0.02 | −16.6 | −3.7 | −0.08 |
| Malan | 9 | G090796E35893N | 5.3 | −0.0 | −0.8 | −0.03 | −0.0 | −0.7 | −0.06 | −0.1 | −1.5 | −0.04 | −0.1 | −1.2 | −0.25 | −0.1 | −2.8 | −0.06 |
| | 10 | G090781E35848N | 10.5 | −0.4 | −3.9 | −0.13 | −0.0 | −0.0 | −0.00 | −0.4 | −4.0 | −0.09 | −0.0 | −0.3 | −0.07 | −0.4 | −4.4 | −0.09 |
| | 11 | G090668E35840N | 10.9 | −0.3 | −2.9 | −0.10 | −0.1 | −1.2 | −0.09 | −0.4 | −4.3 | −0.10 | −0.0 | −0.2 | −0.03 | −0.5 | −4.4 | −0.09 |
| | 12 | G090621E35846N | 12.1 | −0.4 | −3.3 | −0.11 | −0.1 | −0.7 | −0.06 | −0.5 | −4.2 | −0.09 | −0.0 | −0.4 | −0.07 | −0.5 | −4.5 | −0.09 |
| | 13 | G090782E35805N | 23.0 | −0.4 | −1.8 | −0.06 | −0.3 | −1.1 | −0.09 | −0.7 | −3.0 | −0.07 | −0.2 | −0.9 | −0.17 | −0.9 | −3.8 | −0.08 |
| | 14 | G090693E35807N | 32.2 | −1.7 | −5.1 | −0.18 | 0.2 | 0.6 | 0.05 | −1.5 | −4.8 | −0.11 | 0.0 | 0.0 | 0.00 | −1.5 | −4.8 | −0.10 |



|  |  |  |  |  |  |  |  |  |  |  |  |  |  |  |  |  |  |
|---|---|---|---|---|---|---|---|---|---|---|---|---|---|---|---|---|---|
| 15 | G090633E35808N | 14.5 | −0.2 | −1.3 | −0.04 | −0.1 | −0.5 | −0.04 | −0.3 | −1.8 | −0.04 | −0.0 | −0.2 | −0.03 | −0.3 | −2.0 | −0.04 |
| 16 | G090575E35839N | 9.2 | −0.2 | −2.2 | −0.07 | −0.0 | −0.5 | −0.04 | −0.3 | −2.8 | −0.06 | −0.0 | −0.1 | −0.02 | −0.3 | −2.9 | −0.06 |
| subtotal | | 198.2 | −7.9 | −4.0 | −0.14 | −2.2 | −1.2 | −0.09 | −10.1 | −5.1 | −0.12 | −0.7 | −0.4 | −0.07 | −10.8 | −5.4 | −0.11 |
| total | | 641.2 | −24.8 | −3.9 | −0.13 | −2.3 | −0.4 | −0.03 | −27.2 | −4.2 | −0.10 | −0.2 | −0.0 | −0.01 | −27.4 | −4.3 | −0.09 |



**Table 3**. Glacier length ($L$) and changes ($\Delta L$) at Xinqingfeng and Malan for selected glaciers.

| Region | ID | GLIMS ID | $L_{1970/71}$ (km) | $\Delta L_{1970/71-2000}$ (m) | $\Delta L_{2000-2013}$ (m) | $\Delta L_{2013-2018}$ (m) | $\Delta L_{1970/71-2018}$ (m) |
|---|---|---|---|---|---|---|---|
| Xinqingfeng | 1 | G090837E36060N | 12.05 ± 0.01 | 329.3 ± 20.2 | 0.0 ± 16.8 | −83.2 ± 10.6 | 246.1 ± 15.4 |
| | 2 | G090884E36076N | 15.36 ± 0.01 | 584.8 ± 20.2 | 0.0 ± 16.8 | 0.0 ±10.6 | 584.8 ± 15.4 |
| | 3 | G091076E36106N | 9.67 ± 0.01 | −52.9 ± 20.2 | 0.0 ± 16.8 | 0.0 ± 10.6 | −52.9 ± 15.4 |
| | 4 | G091032E36060N | 20.98 ± 0.01 | −2546.8 ± 20.2 | 650.5 ± 16.8 | 513.5 ± 10.6 | −1382.8 ± 15.4 |
| | 5 | G090983E36018N | 9.50 ± 0.01 | −169.1 ± 20.2 | −55.0 ± 16.8 | −216.4 ± 10.6 | −440.5 ± 15.4 |
| | 6 | G090901E36002N | 10.80 ± 0.01 | −45.0 ± 20.2 | −46.8 ± 16.8 | 46.0 ± 10.6 | −45.8 ± 15.4 |
| | 7 | G090868E35998N | 6.90 ± 0.01 | 40.8 ± 20.2 | 107.8 ± 16.8 | −43.4 ± 10.6 | 105.2 ± 15.4 |
| | 8 | G090846E36001N | 5.78 ± 0.01 | 432.0 ± 20.2 | −19.7 ± 16.8 | −75.7 ± 10.6 | 336.6 ± 15.4 |
| | | Selected glaciers (mean) | | −178.4 ± 20.2 | 79.6 ± 16.8 | 17.6 ± 10.6 | −81.2 ± 15.4 |
| | | Selected glaciers (mean annual) | | −5.9 ± 0.7 | 6.1 ± 1.3 | 3.5 ± 2.1 | −1.7 ± 0.3 |
| Malan | 9 | G090796E35893N | 3.87 ± 0.01 | 0 ± 20.2 | −44.5 ± 16.8 | −37.8 ± 10.6 | −82.3 ± 15.4 |
| | 10 | G090781E35848N | 5.20 ± 0.01 | −89.3 ± 20.2 | −17.6 ± 16.8 | −9.7 ± 10.6 | −116.6 ± 15.4 |
| | 11 | G090668E35840N | 5.13 ± 0.01 | −121.6 ± 20.2 | −306.2 ± 16.8 | −78.1 ± 10.6 | −505.9 ± 15.4 |
| | 12 | G090621E35846N | 6.01 ± 0.01 | −220.5 ± 20.2 | −75.4 ± 16.8 | −51.6 ± 10.6 | −347.5 ± 15.4 |
| | 13 | G090782E35805N | 8.61 ± 0.01 | −55.5 ± 20.2 | −82.8 ± 16.8 | −26.1 ± 10.6 | −164.4 ± 15.4 |
| | 14 | G090693E35807N | 9.18 ± 0.01 | −873.9 ± 20.2 | 183.4 ± 16.8 | 0.0 ± 10.6 | −690.5 ± 15.4 |
| | 15 | G090633E35808N | 5.81 ± 0.01 | −60.0 ± 20.2 | −72.7 ± 16.8 | −54.9 ± 10.6 | −187.6 ± 15.4 |
| | 16 | G090575E35839N | 4.28 ± 0.01 | −10.9 ± 20.2 | −35.6 ± 16.8 | −14.8 ± 10.6 | −61.3 ± 15.4 |
| | | Selected glaciers (mean) | | −179.0 ± 20.2 | −56.4 ± 16.8 | −56.4 ± 10.6 | −269.5 ± 15.4 |
| | | Selected glaciers (mean annual) | | −6.0 ± 0.7 | −4.3 ± 1.3 | −11.3 ± 2.1 | −5.6 ± 0.3 |
| Total | | Selected glaciers (mean) | | −178.7 ± 20.2 | 11.6 ± 16.8 | −8.3 ± 10.6 | −175.3 ± 15.4 |
| | | Selected glaciers (mean annual) | | −6.0 ± 0.7 | 0.9 ± 1.3 | −1.7 ± 2.1 | −3.7 ± 0.3 |



**Table 4.** Glacier mean elevation (ΔH) and geodetic glacier mass balance rates measured from DEM differencing.

| Region | ID | GLIMS ID | 1970/71–1999 | | 1999–2011/12 | | 1970/71–2011/12 | | 2000–2016* |
|---|---|---|---|---|---|---|---|---|---|
| | | | Mean ΔH (m) | Annual mass balance (m w.e. a⁻¹) | Mean ΔH (m) | Annual mass balance (m w.e. a⁻¹) | Mean ΔH (m) | Annual mass balance (m w.e. a⁻¹) | Annual mass balance (m w.e. a⁻¹) |
| Xinqingfeng | 1 | G090837E36060N | | | $-4.41 \pm 2.26$ | $-0.29 \pm 0.18$ | | | $-0.22$ |
| | 2 | G090884E36076N | | | $-5.97 \pm 2.32$ | $-0.39 \pm 0.18$ | | | $-0.33$ |
| | 3 | G091076E36106N | $-4.44 \pm 2.75$ | $-0.13 \pm 0.09$ | $-4.41 \pm 2.00$ | $-0.31 \pm 0.15$ | $-8.65 \pm 0.91$ | $-0.17 \pm 0.02$ | $-0.40$ |
| | 4 | G091032E36060N | | | $-4.77 \pm 2.16$ | $-0.34 \pm 0.17$ | | | $-0.02$ |
| | 5 | G090983E36018N | | | $1.43 \pm 2.11$ | $0.10 \pm 0.16$ | | | $0.04$ |
| | 6 | G090901E36002N | | | $1.02 \pm 2.12$ | $0.07 \pm 0.16$ | | | $0.05$ |
| | 7 | G090868E35998N | | | $0.15 \pm 2.09$ | $0.01 \pm 0.16$ | | | $0.02$ |
| | 8 | G090846E36001N | | | $-0.35 \pm 2.00$ | $-0.02 \pm 0.15$ | | | $0.04$ |
| subtotal | | | | | $-3.50 \pm 2.17$ | $-0.22 \pm 0.17$ | | | $-0.14$ |
| Malan | 9 | G090796E35893N | $-3.21 \pm 3.71$ | $-0.09 \pm 0.13$ | $-4.41 \pm 2.15$ | $-0.29 \pm 0.17$ | $-7.85 \pm 0.84$ | $-0.16 \pm 0.02$ | $-0.20$ |
| | 10 | G090781E35848N | $-7.17 \pm 3.48$ | $-0.21 \pm 0.12$ | $-3.96 \pm 2.11$ | $-0.26 \pm 0.17$ | $-10.40 \pm 0.85$ | $-0.21 \pm 0.02$ | $-0.25$ |
| | 11 | G090668E35840N | $-8.00 \pm 3.90$ | $-0.23 \pm 0.13$ | $-2.79 \pm 2.25$ | $-0.18 \pm 0.18$ | $-10.85 \pm 0.81$ | $-0.22 \pm 0.02$ | $-0.08$ |
| | 12 | G090621E35846N | $-12.37 \pm 4.28$ | $-0.36 \pm 0.15$ | $-4.72 \pm 2.24$ | $-0.31 \pm 0.18$ | $-18.65 \pm 0.81$ | $-0.38 \pm 0.02$ | $-0.27$ |
| | 13 | G090782E35805N | $-16.41 \pm 3.89$ | $-0.48 \pm 0.13$ | $-5.22 \pm 2.23$ | $-0.34 \pm 0.18$ | $-20.99 \pm 0.83$ | $-0.42 \pm 0.02$ | $-0.39$ |
| | 14 | G090693E35807N | $0.84 \pm 3.96$ | $0.02 \pm 0.14$ | $-4.36 \pm 2.25$ | $-0.29 \pm 0.18$ | $-3.35 \pm 0.81$ | $-0.07 \pm 0.02$ | $-0.17$ |
| | 15 | G090633E35808N | $0.04 \pm 4.49$ | $0.00 \pm 0.15$ | $-5.26 \pm 2.33$ | $-0.34 \pm 0.18$ | $-5.71 \pm 0.80$ | $-0.12 \pm 0.02$ | $-0.14$ |
| | 16 | G090575E35839N | $-9.46 \pm 3.86$ | $-0.28 \pm 0.13$ | $-5.92 \pm 2.12$ | $-0.39 \pm 0.17$ | $-13.47 \pm 0.84$ | $-0.27 \pm 0.02$ | $-0.39$ |
| subtotal | | | $-6.53 \pm 3.95$ | $-0.19 \pm 0.14$ | $-4.42 \pm 2.21$ | $-0.29 \pm 0.17$ | $-10.72 \pm 0.91$ | $-0.22 \pm 0.02$ | $-0.23$ |
| total | | | | | $-3.78 \pm 2.18$ | $-0.24 \pm 0.17$ | | | $-0.17$ |

*These results were derived from Brun et al. (2017).



**Table 5**. Main characteristics of surged or advanced glaciers.

| Region | ID | GLIMS ID | Advance period | Advance (m) | Advance rate (m a$^{-1}$) | Relatiive length change(%) | Mean slope(°) | Aspect |
|---|---|---|---|---|---|---|---|---|
| Xinqingfeng | 1 | G090837E36060N | 1971-1987 | 278.4 | 17.4 | 2.3 | 13.5 | N |
| | | | 1989-1999 | 50.9 | 5.1 | 0.4 | | |
| | 2 | G090884E36076N | 1987-1989 | 1200.0 | 600.0 | 7.8 | 9.2 | NW |
| | | | 1990-1998 | 256.0 | 32.0 | 1.7 | | |
| | 4 | G091032E36060N | 2009-2016 | 1164.0 | 194.0 | 5.5 | 10.1 | E |
| | 6 | G090901E36002N | 2014-2016 | 46.0 | 23.0 | 0.4 | 17.7 | S |
| | 7 | G090868E35998N | 1986-1989 | 40.8 | 13.6 | 0.6 | 21.5 | S |
| | | | 2009-2010 | 107.8 | 107.8 | 1.6 | | |
| | 8 | G090846E36001N | 1970-1986 | 663.0 | 41.4 | 11.5 | 21.9 | S |
| Malan | 14 | G090693E35807N | 2007-2012 | 260.5 | 52.1 | 2.8 | 7.6 | S |