# Peer review of "Glacier changes and surges over Xinqingfeng and Malan Ice Caps in the inner Tibetan Plateau since 1970 derived from Remote Sensing Data"

_The Cryosphere, 2019_

## Referee Comment (RC1) · Anonymous Referee #1 · 19 Jun 2019

In this article Zhang and colleagues document glacier changes of Xinqingfeng and Malan ice caps for the period 1970-2000 and 2000-2011/12. Using a combination of radar and optical sensors, they investigate glacier mass and area changes of approximately 640 km2 of glaciers. The novelty of this study relies on: 1- to document glacier mass changes for the period 1970-2000 for the Malan ice caps and 2- to report seven previously undocumented glacier surges. The authors investigate the influence of the glacier aspect on the pattern of mass change and discuss the potential mechanisms triggering the surges.

While the data presented in this article are of interest, they cover a limited area (just

as a reference, Zhou et al. (2018) reported glacier mass changes for more than 8 800 km2 of ice for the same period 1970-2000 in a single article). Additionally, there are numerous imprecisions and paragraphs with limited scientific interest in this article, which weaken the article's potential to attract a certain audience. The text is sometimes hard to understand due to severe grammatical issues. The article has also methodological issues, which need to be clarified. The level of precision is often not well chosen: for example, very precise numbers are quoted in the introduction, where the general context is expected, and very few details are provided in the data and methods sections, where they are expected. In the results and discussion sections, I suspect that the authors discuss differences that are not significant because of overlapping error bars. Overall, this paper is far from publication, it needs to be completely rewritten in a more concise and precise way. These issues need to be carefully addressed before resubmission.

Major comments: 1- In my opinion, this article needs to find a real scientific question to be answered, instead of reporting scattered observations. What is the goal of this study? Is it to document long time series of surge activity? Is it to document mass balance changes from 1970-2000 to 2000-2012 in a previously undocumented region? Without a clear focus, the paper reads as a fact report and not as a scientific paper. The introduction has to be completely rewritten, in order to lead the reader to the scientific question addressed by the article. 2- The authors should better justify the choice of the remote sensing data they used in this study. For instance, what is the interest of the GoLive data, which cover only the period 2013-2018, when the authors investigate longer term changes? Is there a way to extend this time series by adding Landsat imagery? 3- Serious methodological concerns: a- About the topographic maps: these specific maps are not commonly for geodetic mass balance, and they are also not publicly available. Consequently, the authors need to demonstrate in a quantitative way the suitability of these data to investigate glacier area and elevation changes. The minimum expected details are: off-glacier map of elevation changes, percentage of voids, parameters used for the transformation methods (and how many GCPs where used? Which residuals?). I am very surprised to see almost complete maps over Malang

ice cap, when ASTER data on fig. 5d show large voids, pointing towards low contrast surfaces that were likely challenging to map for the aerial survey of 1970 as well. The authors need to discuss this issue. b- About the gap filling method/outlier filtering: the authors do not explicitly write how they filter outliers and/or how do they deal with incomplete maps of elevation changes when calculating the mean elevation changes. See McNabb et al. (2019). c- The glacier area changes are not taken into account into the glacier mass balance calculation. As the area changes are limited, I do not expect large differences compared with the current estimate, but it is worth calculating the mass balance in the best possible way. See Fischer et al. (2015) for a methodological description. d- Seasonality and seasonality correction: the climate regime is not described with enough details, and it is not very clear. When are the accumulation and ablation seasons? The TanDEM-X/TerraSAR-X/SRTM data were acquired in Feb-April, whereas the TOPO_DEM data were acquired in Oct-Dec. The authors need to correct for this difference and/or justify why the do not do it. e- SRTM penetration correction. The X-band penetration depth is likely not negligible (unlike it is stated on L11 P4), but the difference between the C and X band penetrations can be approximated by the method described in the article. However, this is an underestimation of the C band penetration, which likely biases the 1970-2000 estimates. This should be kept in mind when comparing the two periods 4- Uncertainty analysis. Eq 3 is not really clear to me. If the corrections of Nuth and Kääb (2011) and Gardelle et al. (2012) are applied Emed should be zero by construction (otherwise it would mean that there is an offset between the two DEMs). Additionally, on P6L29, the authors wrote "of the stable region for each altitude band (50 m)". If they calculate Emed and $\sigma$ for each 50 m elevation band, which value is retained in eq. 3? Similarly, the way Neff is calculated is not very clear. As the glaciers have very different size in their study area, I expect a much wider range of values for the uncertainties as the later depend strongly on Neff (which is proportional to the glacier area) than the values reported in Table 4. These values need to be carefully checked and the authors should provide a range for Neff. P7L21-23: why is the uncertainty in area changes always $\pm 0.03\%a-1$? Please check

the calculation carefully, as this uncertainty should depend both on the glacier area change and the time period considered. 5- Comparison of changes between glaciers with different aspects and between the different periods. In the results and discussion, the authors report differences between the different categories of glaciers, but they should make sure that the error bars do not overlap. They should keep in mind that they investigate only a small sample of glaciers. Specific comments: P1L14: "heterogeneous variations" -> meaning not clear to me P1L15: "there are limited processes available to understand" -> not clear to me. Do you mean that there are no explanation about the heterogeneity? P1L20: "there was" -> "had" P1L18-21: this sentence should be split into two sentences P1L21-22: is this sentence useful in the abstract? P2L3: I would say that the presence of a "Karakoram anomaly" is not much debated. However, you can insist on the fact that its extent is not completely clear, in particular in the inner Tibetan Plateau. P2L4-14: this paragraph is hard to follow because it mixes different results from specific areas, consider simplifying. P2L23 and everywhere else: "surged glaciers" -> "surging glaciers" P3L4: the goal of this study should be stated here in a clear way P3L9: "terminal" -> "terminus" P3L16-20: the climate setting description is too short and cannot be based only on one field campaign of two month P3L27: why is the J-46-134 different from the others P4L2: more details needed on the quality assessment of these maps P4L16: what is the average penetration value? P5L16-17: GCPs were used? P5L26: I don't understand "changing" P6L20: this is in contradiction with paragraph 3.3. How was generated the TSX/TDX-SRTM difference in the end? P6L29: give values of sigma for each DEM difference. P7L2: 30 m is inconsistent with the value of 12 m from P5L11 P7L5: I do not understand the method here P7L8-9: the radar penetration is a systematic bias. Here it is treated as a random error, which is not ok. P7L10: "km" -> kg P7L11: Bolch et al. 2017 -> Huss 2013 P7L16-17: repetition from section 2 P7L21-24: the P7L22: add "insignificantly" before "decreased" P7L24 and everywhere in the text: "advanced" -> "advancing" P7L25: carefully check that the differences are significant before interpreting P8L1-6: are these differences significant? P8L8 and everywhere: inconsistency between "decrease" and "-3.50". I

suggest to stick to "elevation changes" and "mass changes" to avoid these inconsistencies. P8L18-19: are the mass balances significantly different? P8L19: "different slopes" -> "different aspects" P8L24 and after: could you convert the velocities in m a-1? P8L24-27: Here there is the same problem as for the mass balance calculation from the rate of elevation change maps. The velocity maps are likely incomplete and the authors need to explain how they can calculate the average velocity from these maps. P8L28: uncertainty? P9L3-11: Maybe I missed something, but I am not so sure that the arguments of the authors convince me that Zu Glacier is a surge type glacier... The 2013-2018 velocities are very stable, and the front position as well. I agree that the shape of the terminus suggests a surge, but the surge cannot be demonstrated from the data presented here. On figure 9c the authors should plot rate of elevation change instead of elevation changes as the time spans are different. P9L13 and elsewhere: provide a number of digit that reflects the uncertainty "278.4 ± 21.2 m" -> "280 ± 20 m" P9L26-27: the sentence is hard to understand, at the moment it reads as if the glacier retreated due to the lack of data! P9L27-29: it is hard to know if these thinning and thickening are "normal" or "surge-type" behavior, as they do not show the typical surge signature. P10L2: "We confirmed that Glacier No. 14 surged as a consequence of its advance from 2007-2012" -> I don't follow the reasoning here P10L4: not clear. A precise reference to the classification is needed here. P10L19: this statement does not seem back up by much theoretical background. Consider either elaborating or removing. P10L29: about the comparison between the authors estimate and Gardner et al. (2013), the SRTM C band penetration cannot be the explanation of the difference. The authors' estimate should be biased towards too positive values because of the penetration (SRTM is used as a start date for the geodetic estimate), but the opposite is observed with Gardner et al.'s value being less negative. P11L2: "Their estimated trend is more significant than ours" -> "Their estimated trend is significantly more negative than ours" P11L15: I don't understand the reasoning about the turning point P11L18-19 and figure 10: this figure does not demonstrate the authors' point, as almost all glaciers have steeper accumulation area than ablation area. If they want

to reinforce their argumentation they need to show the profiles of retreating glaciers as well to show a difference between these glaciers and the surging and/or advancing ones. Section 5.3: all this section is hard to understand. The authors try to relate general statements to their local observations, but they lack precision and quantification (e.g. "These glaciers have relatively narrow width, so the mass transported from the reservoir area might be limited." -> what to conclude from this? Or "the observed small ice mass losses have not been enough to result in a strong negative mass balance"). In general, I have the feeling that the references, theories and mechanisms discussed are a bit outdated. I suggest to rely on more recent literature and approaches to completely revise this section (e.g., Gilbert et al., 2018; Sevestre et al., 2015). P14L2-3: this statement is extremely speculative P14L16: "decreased" is it significant? If yes, write it, if not remove. P14L19: "lower" than what? P14L27-30: I do not think that the data presented here support this conclusion Figure 2, 4, 6, 7: I do not think that rose diagrams are relevant representation of the data for such a small sample of glaciers. In particular for figures 4, 6 and 7, how are represented the glaciers with positive area or mass changes? Figure 3: the quality of the figure is very poor, it needs to be improved. Figure 5: off-glacier elevation changes should be shown as a supplementary figure. Figure 8 and 9: elevation differences need to be converted into rates of elevation difference (panels c). The transect should be plot on the d panels.

Fischer, M., Huss, M., Hoelzle, M., 2015. Surface elevation and mass changes of all Swiss glaciers 1980–2010. The Cryosphere 9, 525–540. https://doi.org/10.5194/tc-9-525-2015 Gardelle, J., Berthier, E., Arnaud, Y., 2012. Impact of resolution and radar penetration on glacier elevation changes computed from DEM differencing. J. Glaciol. 58, 419–422. https://doi.org/doi:10.3189/2012JoG11J175 Gardner, A.S., Moholdt, G., Cogley, J.G., Wouters, B., Arendt, A.A., Wahr, J., Berthier, E., Hock, R., Pfeffer, W.T., Kaser, G., Ligtenberg, S.R.M., Bolch, T., Sharp, M.J., Hagen, J.O., van den Broeke, M.R., Paul, F., 2013. A Reconciled Estimate of Glacier Contributions to Sea Level Rise: 2003 to 2009. Science 340, 852–857. https://doi.org/10.1126/science.1234532 Gilbert, A., Leinss, S., Kargel, J., Kääb, A., Gascoin, S., Leonard, G., Berthier, E.,

Karki, A., Yao, T., 2018. Mechanisms leading to the 2016 giant twin glacier collapses, Aru Range, Tibet. The Cryosphere 12, 2883–2900. https://doi.org/10.5194/tc-12-2883-2018 McNabb, R., Nuth, C., Kääb, A., Girod, L., 2019. Sensitivity of glacier volume change estimation to DEM void interpolation. The Cryosphere 13, 895–910. https://doi.org/10.5194/tc-13-895-2019 Nuth, C., Kääb, A., 2011. Co-registration and bias corrections of satellite elevation data sets for quantifying glacier thickness change. The Cryosphere 5, 271–290. https://doi.org/10.5194/tc-5-271-2011 Sevestre, H., Benn, D.I., Hulton, N.R.J., Bælum, K., 2015. Thermal structure of Svalbard glaciers and implications for thermal switch models of glacier surging. J. Geophys. Res. Earth Surf. 120, 2220–2236. https://doi.org/10.1002/2015JF003517 Zhou, Y., Li, Z., Li, J., Zhao, R., Ding, X., 2018. Glacier mass balance in the Qinghai–Tibet Plateau and its surroundings from the mid-1970s to 2000 based on Hexagon KH-9 and SRTM DEMs. Remote Sens. Environ. 210, 96–112. https://doi.org/10.1016/j.rse.2018.03.020

---

## Referee Comment (RC2) · Anonymous Referee #2 · 22 Jun 2019

In this article, Zhang et al. document glacier changes in the Xinqinfeng and Laman Ice cap, inner Tibetan plateau, from 1970 to 2018 from means of various remote sensing data. They examine glacier area changes, mass changes and velocity of the glaciers and report several glacier surges during the period of observation. While the observation have a certain value in a previously poorly documented region, the analysis and interpretation lack clarity and objectives, which makes the reading of the article extremely difficult.

Major comments:

In general, I agree with the comments provided by referee #1 and in particular:

[Figure]

1. The article lacks a scientific question and a logic throughout the text. Currently it is essentially a report of observations and the scientific value of these observations is lost in a lot of details. The structure of the paper should be revised and the text significantly reduced in order to provide a concise and clear message. Most numbers discussed in the text are purely informative and should be summarized in tables, if really important to answer the scientific question of the article, rather than enumerated. Section 5.3 (discussion) suddenly contains methods (around p 10-L12-17) and results (p9-L18 until the end of paragraph) that should be moved to the appropriate section and introduced.

2. The whole discussion about the dependence of the area/mass changes with topographic characteristics (essentially aspect) seem anecdotical. The sample size is relatively small (77 and 59 glaciers) and once divided into 8 aspects represents less than 10 glaciers per sample on average. All the spatial variability is basically dominated by individual glaciers behavior and in this sense the discussion does have much scientific interest. I recommend to remove this discussion along with figures 2,4, 6 and 7 to focus on other more valuable aspects of the paper.

3. The justification of the data and methods used is often unclear. The authors try to provide a picture as complete as possible of the changes affecting these glaciers but this gives a general impression of scattering and not enough exploitation of the available data and in-depth analysis. For example, why focus on two dates of ASTER data when 19 years are available. What do the generated results bring compared to the observations of Brun et al (2017), who provide at least two time period (2000-2008 and 2008-2016)? Similarly, the analysis of the glacier velocities lacks some depth. This is of course partly due to the fact that GoLIVE observation are only available since 2013, but over such a limited area, the exploitation of Landsat data to look at velocities over the entire time period would be interesting. Alternatively, regional datasets are and/or will soon be available (see for example Dehecq et al., 2019). Moreover, if I recall correctly the GoLIVE data are generated from 16-64 days image pairs. How is

the annual velocity showed in figures 8 and 9 estimated? Is this just one velocity field or an average of all pairs in a year? What about the seasonal variability? What is the uncertainty of the observations? Please explain.

4. The elevation change results raise some concern. First, while the C-band radar penetration is discussed, the X-band penetration is ruled out very quickly. The assumption that X-band penetration is negligible (made for example in Gardelle et al., 2012a referenced in the study), has since then been quite criticized and X-band penetration in snow/ice has been shown to reach several meters in high altitude and dry conditions (Leinss et al. 2015; Dehecq et al. 2016, Abdullahi et al., 2018), which is most likely the case in your study area in February (SRTM-X) or March (TanDEM-X). A penetration of several meters is not negligible over a period of 2-3 years as represented on your figures 8c and 9c and also similar to your estimated C-band penetration (Figure S1). This should be taken into account in the uncertainty estimate and the interpretation of these results. Second, the elevation changes shown on figures 8 and 9 show a very large spatial variability with an amplitude over 100 m and an elevation gain in the glacier accumulation zone (blue curve panel 8c) that are highly suspicious. The time periods of a few years discussed seem too small compared to the apparent uncertainty of the observations. Third, the elevation change maps on Figure 5 show some suspicious patterns of elevation gain/loss particularly in Malan and for the historical period. As pointed out by referee #1, more quantitative measurements of the suitability of these data for elevation change analysis (off ice statistics, distribution with altitude/slope etc, see for example Gardelle et al 2012 a/b or Girod et al 2017) must be provided and the uncertainty related to the seasonality of the observation must be discussed (Gardelle et al 2013).

Minor comments:

- p1-L14: "with heterogenous variations" -> be more specific

- p1-L19: I think a "per year" is missing in the area change values

- p1-L31: "Bhutan, Nepal and Spiti-Lahaul" are part of "the Himalayas", so both groups should not be considered as separate.

- p2-L16: "glacier balance heterogeneous sub-regions" -> This part does not make sense. Maybe "aggregation of sub-regions with heterogeneous glacier balance conditions"?

- p6-L20-21 : The reference to Nuth & Kaab should probably be after "altimetric shift" rather than "differences between the DEMs".

- p7-L27: Until the end of paragraph. This section should probably be removed (see major comment 2)

- p8-L19-23: Same here.

- p8-L24-27: What is the interest of a mountain range average velocity? You bring up the difference in slope to explain the difference in average velocity, but what about the ice thickness? This paragraph says either too much or too little.

- p8-L30: the time periods mentioned in the text don't match the time periods on the figures. Please explain or rephrase.

- p9-L3-9: The variability and uncertainty of the velocities are huge compared to the signal. Similarly, for the elevation change (figure 9). I really don't find that the results provide sufficient evidence for a surge. Please provide better evidence of it or remove this paragraph.

- Figure 8 and 9: If I recall correctly, GoLIVE velocities are derived from 16-64 days image pairs. How do you estimate your annual velocity? Is each curve extracted from just one pair or an average of several pairs? If so, which ones? Please explain.

- p9-L12-20: this whole paragraph is redundant with the Table and could be much more concise.

- Table 2: Remove column 1970-2013 as it is essentially redundant with the other

results. In general, the tables 2-4 could probably be reduced to the most important information.

- Fig 2: What do the numbers on the radial axis mean?

- Fig 5: The legend font should be increased a lot! Maybe by combining all legends into one. The color scale should be improved, maybe by using a logarithmic scale, as a change of +/-4 m/yr is huge and the scale masks probably large noise in the accumulation areas.

- throughout the text: surged glacier -> surging glacier; advanced glacier-> advancing glacier

Additional references:

Abdullahi, S., Wessel, B., Leichtle, T., Huber, M., Wohlfart, C., Roth, A., 2018. Investigation of Tandem-x Penetration Depth Over the Greenland Ice Sheet, in: IGARSS 2018 - 2018 IEEE International Geoscience and Remote Sensing Symposium. Presented at the IGARSS 2018 - 2018 IEEE International Geoscience and Remote Sensing Symposium, pp. 1336–1339. https://doi.org/10.1109/IGARSS.2018.8518930

Dehecq, A., Gourmelen, N., Gardner, A.S., Brun, F., Goldberg, D., Nienow, P.W., Berthier, E., Vincent, C., Wagnon, P., Trouvé, E., 2019. Twenty-first century glacier slowdown driven by mass loss in High Mountain Asia. Nature Geoscience 12, 22. https://doi.org/10.1038/s41561-018-0271-9

Dehecq, A., Millan, R., Berthier, E., Gourmelen, N., Trouve, E., Vionnet, V., 2016. Elevation Changes Inferred From TanDEM-X Data Over the Mont-Blanc Area: Impact of the X-Band Interferometric Bias. IEEE Journal of Selected Topics in Applied Earth Observations and Remote Sensing 9, 3870–3882. https://doi.org/10.1109/JSTARS.2016.2581482

Girod, L., Nuth, C., Kääb, A., McNabb, R., Galland, O., 2017. MMASTER: Improved ASTER DEMs for Elevation Change Monitoring. Remote Sensing 9, 704.

https://doi.org/10.3390/rs9070704

Leinss, S., Wiesmann, A., Lemmetyinen, J., Hajnsek, I., 2015. Snow Water Equivalent of Dry Snow Measured by Differential Interferometry. IEEE Journal of Selected Topics in Applied Earth Observations and Remote Sensing 8, 3773–3790. https://doi.org/10.1109/JSTARS.2015.2432031

---

## Author Comment (AC1) · 21 Aug 2019

Please find the final responses to all two comments, the revised manuscript, the revised manuscript with highlighting of changes and the added supplement to the manuscript in the supplement to this author comment.

Please also note the supplement to this comment:
https://www.the-cryosphere-discuss.net/tc-2019-94/tc-2019-94-AC1-supplement.zip